# *Citrus junos* Tanaka Peel Extract Ameliorates HDM-Induced Lung Inflammation and Immune Responses In Vivo

**DOI:** 10.3390/nu14235024

**Published:** 2022-11-25

**Authors:** Dahee Shim, Hwa-Jin Kim, Jungu Lee, You-Min Lee, Jae-Woong Park, Siyoung Yang, Gyeong-Hweon Lee, Myoung Ja Chung, Han-Jung Chae

**Affiliations:** 1Research Institute of Clinical Medicine of Jeonbuk National University-Biomedical Research Institute & Non-Clinical Evaluation Center, Jeonbuk National University Hospital, Jeonju 54907, Republic of Korea; 2Lotte R&D Center, Seoul 07594, Republic of Korea; 3Department of Pathology, Jeonbuk National University Medical School, Jeonju 54907, Republic of Korea; 4School of Pharmacology and Pharmaceutical Research Institute of Korea Unification, Jeonbuk National University, Jeonju 54896, Republic of Korea

**Keywords:** allergic asthma, *Citrus junos* Tanaka, house dust mite, lung inflammation

## Abstract

In the wake of the COVID-19 pandemic, lung disorders have become a major health concern for humans. Allergic asthma is the most prevalent form of asthma, and its treatments target the inflammation process. Despite significant developments in the diagnosis and management of allergic asthma, side effects are a major concern. Additionally, its extreme heterogeneity impedes the efficacy of the majority of treatments. Thus, newer, safer therapeutic substances, such as natural products, are desired. *Citrus junos* Tanaka has traditionally been utilized as an anti-inflammatory, sedative, antipyretic, and antitoxic substance. In this study, the protective effects of *Citrus junos* Tanaka peel extract (B215) against lung inflammation were examined, and efforts were made to understand the underlying protective mechanism using an HDM-induced lung inflammation murine model. The administration of B215 reduced immune cell infiltration in the lungs, plasma IgE levels, airway resistance, mucus hypersecretions, and cytokine production. These favorable effects alleviated HDM-induced lung inflammation by modulating the NF-κB signaling pathway. Hence, B215 might be a promising functional food to treat lung inflammation without adverse effects.

## 1. Introduction

Asthma is associated with allergic reactions and affects about 300 million individuals globally [1]. Allergic asthma is the most prevalent form of asthma, characterized by airway hyperresponsiveness, lung inflammation, and airway remodeling [2]. Previously, mice were challenged with allergens such as ovalbumin, *Aspergillus fumigatus*, and house dust mite (HDM) to investigate the pathophysiology of allergic asthma. Among these, HDM is the most common risk factor associated with the development of asthma in children and adults [2]. Thus, HDM could be a significant risk factor for the development of allergic airway illnesses in mice. Additionally, HDM induces clinically relevant pro-inflammatory cytokines, eosinophils, neutrophils, and airway hyperresponsiveness and remodeling [3]. Hence, HDM-induced mice model was used for the investigation. Generally, asthma patients were treated with corticosteroids and long-acting bronchodilators; nevertheless, these treatments have their own side effects [4]. Moreover, there are no preventive medications for pulmonary inflammation or asthma. Functional foods derived from natural products could be an alternate approach to prevent chronic asthma or lung inflammation without side effects.

*Citrus junos* Tanaka, also known as yuzu, is a citrus fruit mainly cultivated in Japan and Korea and widely used during the preparation of tea and sauce. Traditionally, it is used to treat bronchial and respiratory illnesses [5]. Previous investigations have shown that yuzu is a rich source of flavonoids, including naringin, hesperidin, neohesperidin, and narirutin. Several preclinical studies indicate that naringin has antioxidant, anti-inflammatory, and antiapoptotic properties [6]. Additionally, multiple animal models demonstrated a mitigable effect of yuzu on colitis, hepatic steatosis, obesity, hepatic lipid accumulation, and neurodegeneration [7,8,9,10]. Furthermore, water extracts of yuzu peel alleviated acute lung injury in acrolein [11] or particulate matter-10-induced mouse models [12,13]. In this study, the protective effects of *Citrus junos* Tanaka peel extract (B215) against lung inflammation were examined, and efforts were made to understand the underlying protective mechanism using an HDM-induced lung inflammation murine model.

## 2. Materials and Methods

### 2.1. Preparation and Analysis of Citrus junos Tanaka Peel Extract (B215)

#### 2.1.1. Preparation of B215

*Citrus junos* Tanaka peel extract (B215) was prepared as described previously with minor modifications (35268078, 35684069). Briefly, dried *Citrus junos* Tanaka peels were purchased from Udo Food corporation, Namhae, Korea. The sample specimen (B215) was deposited at the Lotte research and development center (Seoul, Republic of Korea). Generally, citrus peels were extracted in either distilled water or 50% ethanol with variable extraction times (2, 3, and 6 h). In this study, B215 was extracted in 50% ethanol for 3 h. Specifically, dried *C. junos* Tanaka peels were extracted in 50% ethanol (*v*/*v*) at 80 °C for 3 h. The extract was filtered, concentrated under reduced pressure, and freeze-dried. This *Citrus junos* Tanaka peel extract (B215) contains naringin. Additionally, commercially available naringin was purchased from Sigma-Aldrich Chemical Co. (#10236-47-2, St. Louis, MO, USA) for comparative analysis. All the other chemicals and reagents were of high-performance liquid chromatography (HPLC) or ACS grade.

#### 2.1.2. HPLC Analysis

The quality of B215 was determined by the amount of naringin in the extract. For analysis, 0.1 g of the B215 was mixed with 10 mL of methanol. The suspension was sonicated for 10 min and filtered through a 0.45 µm syringe filter prior to injection into the HPLC system. Agilent 1260 infinity II HPLC system (Agilent Technologies, Santa Clara, CA, USA) was equipped with a G7112B binary pump, a G7115A diode array detector (DAD) and G7129C autosampler. The chromatographic separations were performed on an Agilent Zorbax Eclipse Plus C18 column (4.6 mm × 250 mm, i.d., 5 µm) maintained at 25 °C. The mobile phases were 0.5% aqueous acetic acid (*v*/*v*, A), and acetonitrile (*v*/*v*, B). Gradient elution was performed as follows: 0–5 min, 5–18% B; 5–30 min, 18–18% B; 30–31 min, 18–100% B; 31–35 min, 100–100% B; 35–36 min, 100–5% B; 36–40 min, 5–5% B. The flow rate was 1.0 mL/min, and the injection volume was 10 µL. The eluents from the column were monitored at 280 nm for naringin. Each experiment was performed in triplicate.

### 2.2. Animals and Experimental Design

Female C57BL/6 mice aged eight weeks were acquired from Samtako Bio Korea (Osan, Republic of Korea). Mice were housed under a controlled environment (23 ± 3 °C, 55 ± 15% humidity and 12 h day/night cycle) in SPF facility and received water and a standard diet ad libitum. All the animal experiments were performed as described in the previous study with minor modifications [14]. After one week of acclimatization, 100 μg HDM (Sigma, USA) was intratracheally delivered weekly to mice. After 3 weeks, blood samples were collected to determine the plasma IgE levels. HDM-challenged mice had higher IgE concentrations in their plasma than the PBS-challenged mice.

Mice were separated into five groups following confirmation of HDM-induced lung inflammation. Study groups include CTR, control group; HDM, HDM-induced mice treated with PBS; HDM+50B215, HDM-induced mice treated with 50 mg/kg B215; HDM+100B215, HDM-induced mice treated with 100 mg/kg B215; HDM+200B215, HDM-induced mice treated with 200 mg/kg B215. Each group had 13 mice, and the body weight of all the experimental mice were assessed once a week during treatment with or without B215. All the mice were treated with or without B215 orally once a day for 4 weeks. After completion of the treatment, 8 mice were euthanized using ketamine To collect bronchoalveolar lavage fluid (BALF), blood, and lungs for histological and immunological analysis. The remaining 5 mice from each group were sacrificed to compare airway resistance. All the animal procedures were performed as per the guidelines set and approved by the Institutional Animal Care and Use Committee (IACUC) of Jeonbuk National University Hospital (JBUH-IACUC-2022-3).

### 2.3. Histological Analysis

Histological analyses were performed as described previously [14]. In brief, collected lobes were fixed with a neutral 10% formalin solution and embedded in paraffin. These processed paraffin-embedded tissues were sectioned (4 μm), deparaffinized, rehydrated, and stained with hematoxylin and eosin (H&E). After staining, the perivascular and peribronchial inflammatory scores were analyzed using newly revised criteria, where scores were measured on a subjective scale from 0 to 3 as follows; 0, bronchi with no surrounding leukocyte infiltration; 1, few infiltrating leukocytes; 2, from 1 to 2 layers of perivascular and/or peribronchial leukocytes; 3, from 3 to 5 layers of perivascular and/or peribronchial leukocytes [15]. Further, periodic acid-Schiff (PAS) staining was performed on tissue sections to visualize mucus [14].

### 2.4. Plasma Analysis

Plasma samples were obtained by centrifuging blood at 3000 rpm for 15 min and stored at −80 °C. Enzyme-linked immunosorbent assay (ELISA) was performed to determine IgE levels (#88-50460, Invitrogen, Waltham, MA, USA). Additionally, levels of aspartate aminotransferase (AST, #AM-103K), alanine aminotransferase (ALT, #AM-102-K), triglyceride (TG, #AM-157S-K), and cholesterol (CHOL, #AM-202-K) were performed according to the manufacturer’s instruction (Asan Pharmaceuticals, Seoul, Republic of Korea).

### 2.5. Analysis of Airway Resistance to Methacholine Administration

Anesthesia of mice was achieved using ketamine via intraperitoneal injection, and airway responsiveness against methacholine was measured as described earlier [16]. Before tracheostomy, trachea was exposed via mid-cervical incision. Mice were connected to flexiVent (SCIREQ, Montreal, QC, Canada), a computer-controlled ventilation system with an 18-gauge needle. Mice were ventilated quasi-sinusoidally with the nominal tidal volume of 10 mL/kg of body weight at a frequency of 150 breaths/minute and a positive end-expiratory pressure of 2 cm H_2_O, which could achieve a mean lung volume close to that during spontaneous breathing.

In this investigation, methacholine (Sigma, St. Louis, MO, USA) was selected as a bronchoconstriction reagent. Aerosol methacholine was produced in a nebulizer present within the flexiVent system and administrated through the ventilator into the trachea of mice. To determine responsiveness against methacholine, each mouse was challenged with increasing concentrations of methacholine (0, 5, 10, 25, and 50 mg/mL in saline). After each challenge, the value of respiratory system resistance (Rrs) was calculated. Changes in Rrs (cmH_2_O·s/mL) against each concentration of methacholine were presented to evaluate the airway resistance in HDM-challenged mice.

### 2.6. Counting Immune Cells in BALF

After euthanasia, BALF samples were obtained by centrifuging at 1800 rpm for 3 min with sterile normal saline (0.9% NaCl). To assess immune cells in alveoli, pellets from BALF centrifugation were mixed for total cell counts with a Z1 particle counter (Beckman-Coulter, Indianapolis, IN, USA). Cytospin slides were prepared using cells in the BALF and stained with Diff-Quick (Baxter, Chicago, IL, USA). Under light microscopy, immune cell populations, including neutrophils, eosinophils, basophils, macrophages, and lymphocytes, were counted.

### 2.7. Western Blotting

Tissue treatment and immunoblotting were performed as described previously with minor modifications [14]. In brief, collected lungs were washed, homogenized, and lysed with NE-PER Nuclear and Cytoplasmic Extraction Reagents (Thermo, Waltham, MA, USA). Protein samples were separated on 10% SDS-PAGE, transferred onto PVDF membrane, and probed with indicated antibodies. The protein bands were visualized using ECL reagent and quantified by densitometry analysis using ImageJ (National Institutes of Health, Bethesda, MD, USA). Protein expression levels (band intensities) were normalized against β-actin or histone H3 for cytosolic or nuclear fraction, respectively.

### 2.8. Gene Expression Analysis

Total RNA from right superior and inferior lobes was isolated with Trizol reagent (Invitrogen, Waltham, MA, USA). First-strand cDNA was synthesized using PrimeScript RT reagent Kit (Takara, Nojihigashi, Kusatsu, Japan). qPCR was performed to measure the expression of IL-1β, IL-6, and TNF-α. The details of the primers used in the qPCR are listed in Table 1.

### 2.9. Statistical Analysis

Data are expressed as the mean ± SD. GraphPad Prism version 8.0.1 (GraphPad Software Inc., San Diego, CA, USA) was used for all statistical analyses. Student’s *t*-tests and one-way ANOVA with Tukey post hoc were applied to examine the statistical significance of differences between groups. The threshold of *p* < 0.05 was designated as statistically significant for all tests.

## 3. Results

### 3.1. Establishment of B215 Process

Previously, oral administration of B215 was observed to have a strong protective effect against PM10-induced lung inflammation and its associated injury models [12,13]. Additionally, HDM is the strong, stimulant-associated pulmonary hypofunction and the most common risk factor associated with the development of asthma in children and adults. Hence, we used B215 against HDM-induced lung inflammation. In this study, prepared B215 was analyzed by HPLC for naringin, a major component of Citrus junos Tanaka peel. Prepared B215 contained 3.06 ± 0.06 mg/g naringin (Figure 1).

### 3.2. Administration of B215 Alleviated HDM-Induced Lung Inflammation In Vivo

HDM-induced lung inflammation has been demonstrated to involve allergenic immune responses in lungs, indicated by increased IgE concentration in the plasma, immune cell infiltration, mucus secretion by goblet cells, and higher airway resistance [20]. Expectedly, HDM-induced mice showed increased inflammatory cell infiltration in the perivascular and peribronchial area than the normal control group (Figure 2A). However, supplementing 100 and 200 mg/kg of B215 reduces immune cell infiltration into the lung in HDM-induced mice (Figure 2A). Additionally, plasma IgE concentrations were evaluated to ensure the HDM-induced lung inflammation and allergic immune responses. HDM-induced mice revealed elevated levels of plasma IgE (1.8569 ± 0.6456 μg/mL) compared to the control group (0.8982 ± 0.3193 μg/mL). Furthermore, the administration of B215 to HDM-induced mice resulted in a dose-dependent decrease in plasma IgE levels (Figure 2B). Together, these findings indicate that B215 potentially alleviates HDM-induced lung inflammation.

Next, physical and biochemical parameters were analyzed to examine the systemic effect of B215 in vivo (Figure 3 and Table 2). The physical weight of each animal was measured weekly from the beginning to the end of the trial period. HDM-induced mice gained 6.272 ± 2.980%, while the control group gained 11.811 ± 5.788% at the end of the trial period (Figure 3B). B215-administered, HDM-induced mice were found to weigh the same as the control group. However, the differences between the groups are statistically insignificant (Figure 3B). Furthermore, the plasma AST, ALT, TG, and CHOL were measured as indicators of hepatotoxicity. In this study, all the groups had comparable values for these biochemical parameters (Table 2). Collectively, these data indicate that the administration of B215 has no effect on body weight and does not cause hepatoxicity in vivo.

Airway remodeling, excessive mucus production, and airway hyperresponsiveness developed during chronic allergic inflammation [21]. Among these developments, mucus hyperproduction by goblet cells could elicit clinical symptoms, including globus, bronchial obstruction, and dyspnea [22]. Furthermore, the hyperproduction of mucus in lung airway is associated with increased mortality in patients with asthma, chronic obstructive pulmonary disease, and cystic fibrosis [23]. Especially in asthma, the severity of airway hyperresponsiveness is associated with an increased risk of exacerbation and symptoms, which are regarded as key clinical features of asthma [24]. Therefore, mucus production and airway resistance were assessed to determine the effect of B215 on pulmonary inflammation. Initially, mucus secretion in large airway was assessed by PAS staining using lung sections. HDM-induced mice showed increased mucus secretion, whereas B215 supplementation markedly showed reduced mucus secretion (Figure 4).

Airway hyperresponsiveness was assessed using the flexiVent system with serial methacholine exposure to determine airway hyperresponsiveness. Upon the challenge with methacholine, airway resistance gradually increased with increasing methacholine concentration in all the groups (Figure 5A). The highest airway resistance was observed in HDM-induced mice than in the other groups. The administration of B215 dose-dependently lowered airway resistance (Figure 5A,B). Taken together, the administration of B215 potentially prevents clinical symptoms by regulating mucus secretion and airway resistance in vivo.

### 3.3. Pulmonary Immune Responses Were Controlled by Administration of B215 on HDM-Induced Lung Inflammation

It is a well-known fact that HDM potentially elicits pulmonary inflammation characterized by immune cell infiltration into the lung and alveolar space, elevated levels of inflammatory cytokines, and modifications in inflammation-linked cell signaling pathways [25,26,27]. Thus, we examined the effects of B215 on immune responses in HDM-induced mice. Immune cell populations, including neutrophils, eosinophils, basophils, macrophages, and lymphocytes in the BALF of HDM-induced mice, differed significantly from the control group (Table 3). The supplementation of B215 lowered the number of neutrophils, eosinophils, macrophages, and basophils when compared to HDM-induced mice. Similarly, lymphocytes were improved upon B215 supplementation. These observations suggest that the administration of B215 could modulate the infiltration of myeloid immune cell populations to reduce HDM-induced lung inflammation.

Previous investigations demonstrated the involvement of various signaling pathways, such as NF-κB, MAPK, PTEN/PI3K/AKT, and STAT6, in lung inflammatory diseases [28,29,30,31]. NF-κB signaling pathway play a central role in pulmonary inflammation and could be activated in epithelial cells by repeated exposure to allergen [3]. Additionally, NF-κB is highly activated in asthma patients [32]. Activated NF-κB translocate and phosphorylates, which triggers genes associated with pulmonary inflammation [29]. In this study, the lungs of HDM-induce mice exhibited elevated NF-κB phosphorylation in the nucleus compared to other groups (Figure 6A). The administration of B215 significantly reduced the phosphorylated NF-κB in the nucleus (Figure 6A). However, comparable levels of total nuclear NF-κB were observed in all groups (Figure 6B). Furthermore, cytosolic NF-κB was decreased in the lungs of HDM-induced mice, while administration of 200 mg/kg of B215 nearly restored the cytosolic NF-κB (Figure 6D). Similarly, cytosolic IκB decreased in the lungs of HDM-induced mice, and the administration of B215 dose-dependently restored the cytosolic IκB (Figure 6C). These specific observations suggest that B215 alleviates HDM-induced inflammation by modulating the NF-κB signaling pathway.

It has been suggested that immune cells could produce pro-inflammatory cytokines, including IL-6, colony-stimulating factor, IL-25, and IL-33, upon HDM exposure [33]. Additionally, IL-1β, IL-6, and TNF-α could be produced by epithelial cells in response to allergens [34,35]. Therefore, IL-1β, IL-6, and TNF-α gene expressions were examined to evaluate the anti-inflammatory effects of B215 against HDM-induced lung inflammation in vivo. The expressions of IL-1β (*p* = 0.0002; Figure 7A), IL-6 (*p* = 0.0004; Figure 7B), and TNF-α (*p* = 0.0013; Figure 7C) genes were elevated in the lungs of HDM-induced mice compared to the control group. In contrast, the administration of B215 lowered the IL-1β, IL-6, and TNF-α gene expressions, and expression levels were comparable to the control group (Figure 7). Taken together, the administration of B215 alleviated HDM-induced lung inflammation through the modulation of immune responses in vivo.

## 4. Discussion

Airway inflammation is central to the pathophysiology and clinical expression of asthma. The clinical manifestations of asthma include dyspnea, wheezing, chest tightness, cough, and airway obstruction [36]. In general, asthma is expressed in a range of intensities, from mild to severe but only 5% of asthma cases are severe. This heterogeneity of asthma depends on multiple factors, including lifestyle, age, occupation, allergic condition, history and/or presence of sinusitis [36,37]. Further, comorbidities such as rhinosinusitis, gastroesophageal reflux disease, obstructive sleep apnea, obesity, anxiety, and depression increase the risk of frequent exacerbations [36]. Additionally, these comorbidities are often detrimental to asthma treatment. Traditionally, inhaled corticosteroids are widely accepted treatment for allergic asthma. However, ICS is linked to systemic adverse effects such as osteoporosis, bone fractures, diabetes, ocular disorders, and respiratory infections [36]. Hence, functional foods derived from natural products could be a safer and more effective treatment for allergic asthma.

We hypothesis that B215 derived from natural sources rich in flavonoids with antioxidant, anti-inflammatory, and antiapoptotic properties. Hence, B215 could be a good candidate for preventing HDM-induced lung inflammation. In this study, the effects and underlying protective mechanism of B215 were evaluated using HDM-induced lung inflammation murine model since HDM is the most prominent risk factor for developing allergic airway disease. The administration of B215 efficiently inhibited the inflammatory cell infiltration into the lung (Figure 2A) and effectively regulated plasma IgE levels (Figure 2B), mucus hypersecretion (Figure 4), airway resistance (Figure 5), pulmonary function, and immune cell populations (Table 3). These key findings strongly suggest the preventive or regulatory properties of B215 against asthmatic conditions or allergies.

Reactive oxygen species (ROS) are associated with NF-κB signaling and play a critical role in the anti-inflammatory effects of several functional foods rich in isoflavonoids [38,39,40]. Narirutin, a key flavonoid of *Citrus junos*, was reported to have an anti-inflammatory effect on lipopolysaccharide-stimulated macrophages by lowering IL-β, TNF-α, nitric oxide (NO), and prostaglandin E2 through the inhibition of NF-κB and MAPKs activation in vitro [41]. Additionally, administration of narirutin suppressed lung inflammation in ovalbumin-induced mice, as indicated by lower eosinophilia in BALF and plasma IgE levels [42]. More importantly, naringin is indicated to have anti-inflammatory and anti-apoptotic effects against PM10-induced lung injury [13]. Further, B215 rich in naringin demonstrated inhibitory roles in the activation of AKT, ERK, JNK, and NF-κB in PM10-induced lung damage in vivo with reduced NO production [12].

Narirutin and naringin showed ROS scavenging effects [12,13,39]. In this study, the administration of B215 exhibited anti-inflammatory characteristics and reduced the NF-κB signaling affecting inflammation in vivo (Figure 6). In addition, the IKK-IκBα degradation axis is a well-known mechanism that enhances NF-κB activation [43] and modulates NF-κB-mediated transcription of downstream proinflammatory cytokines [44,45,46]. Levels of IL-1β, IL-6, and TNF-α were decreased by B215 administration in HDM-induced lung inflammation in vivo (Figure 7). IL-1β plays a key role in neutrophilic inflammation during viral-induced asthma exacerbations with increased expression of IL-33 [47]. Furthermore, a recent study revealed that the inhibition of IL-6 and TNF-α could assist in the prevention of allergic asthma in a mouse model [48]. Therefore, a strategy that reduces IL-1β, IL-6, and TNF-α levels can control allergic asthma with or without viral infections. This study suggested that the administration of B215 potentially controls innate immune responses by inhibiting NF-κB signaling.

### Limits of the Study

B215 regulatory effect on adaptive immune responses is still unknown. Additionally, it is unknown which B215 components contributed most to the anti-inflammatory activity against HDM-induced lung inflammation despite the fact that naringin is the major component of B215. Thus, in-depth investigations are necessary to validate the regulatory effect and the component responsible for the anti-inflammatory effect.

## 5. Conclusions

This study strongly suggests that the administration of B215 protects against HDM-induced asthma and allergic reactions. The administration of B215 reduced immune cell infiltration in the lungs, plasma IgE levels, airway resistance, mucus hypersecretions, and cytokine production. At a molecular level, B215 alleviates HDM-induced inflammation by modulating the NF-κB signaling pathway. 

## Figures and Tables

**Figure 1 nutrients-14-05024-f001:**
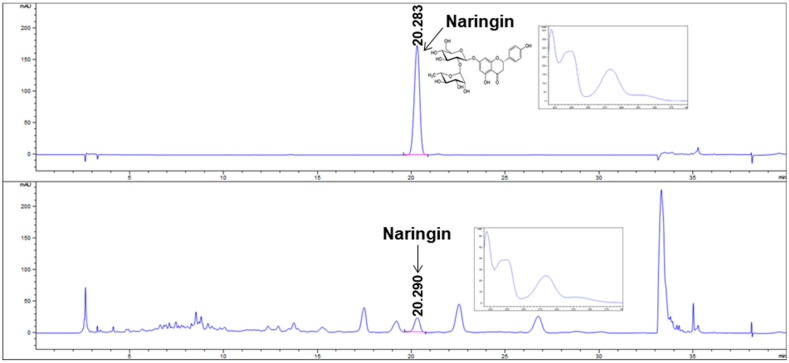
HPLC analysis of naringin and B215. Representative image of HPLC chromatogram at 280 nm of naringin (**upper**) and B215 (**lower**).

**Figure 2 nutrients-14-05024-f002:**
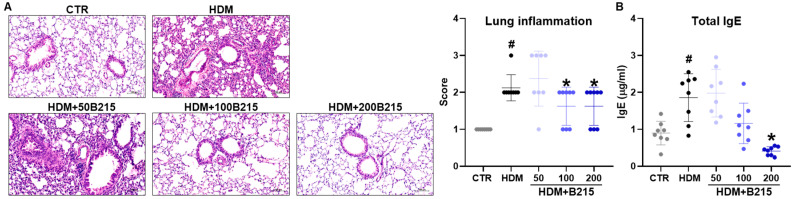
Histological and serological analysis showing the effect of B215 against HDM-induced lung inflammation. (**A**) Representative images of H&E-stained lung sections (Scale bar 100 μm) and inflammatory scores of lungs; (**B**) Plasma IgE levels in different groups. Gray, Control; Black, HDM; Light blue, HDM+50B215; Blue, HDM+100B215; Dark blue, HDM+200B215. Data are presented as mean ± SD (# and *, *p* < 0.05, compared to CTR and HDM, respectively).

**Figure 3 nutrients-14-05024-f003:**
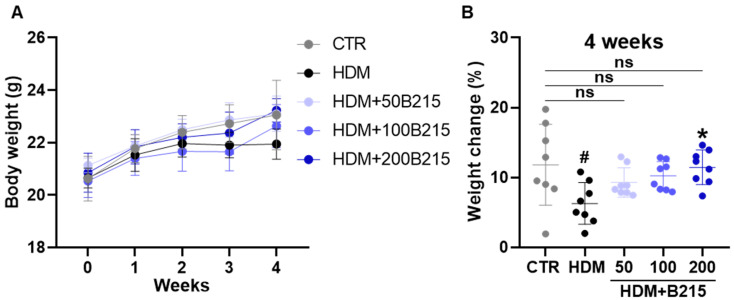
Effect of B215 on body weight in HDM-induced lung inflammation. (**A**) Body weight (**B**) Percentage change in body weight. Gray, Control; Black, HDM; Light blue, HDM+50B215; Blue, HDM+100B215; Dark blue, HDM+200B215. Data are presented as mean ± SD (# and *, *p* < 0.05, compared to CTR and HDM, respectively). ns; not significant.

**Figure 4 nutrients-14-05024-f004:**
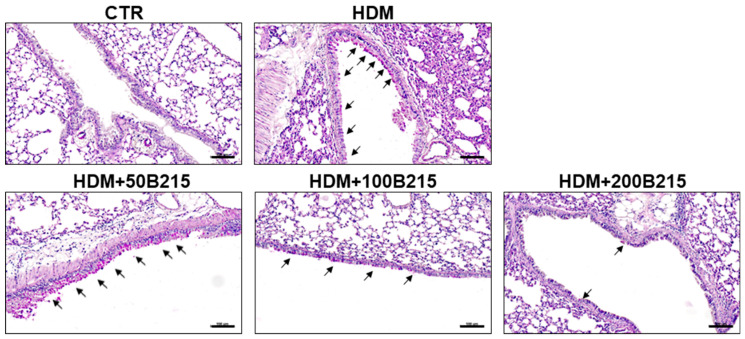
Representative images of PAS-stained lung histology sections. Arrow indicates PAS-stained, mucus-producing goblet cells in large airways (Scale bar 100 μm).

**Figure 5 nutrients-14-05024-f005:**
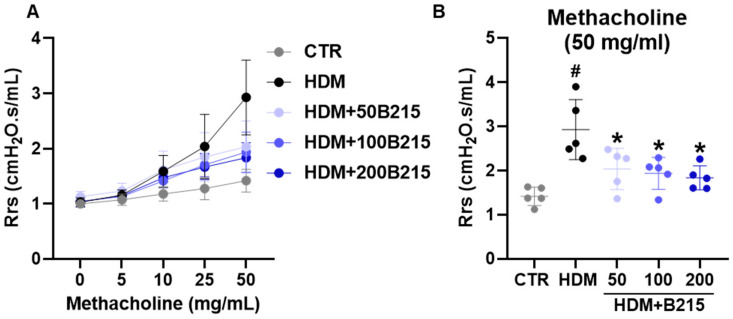
Airway hyperresponsiveness to methacholine in HDM-induced mouse model of lung inflammation. (**A**) Airway responsiveness to aerosolized methacholine (5 to 50 mg/mL) in different groups. (**B**) Airway responsiveness in different groups when challenged with 50 mg/mL methacholine. Gray, Control; Black, HDM; Light blue, HDM+50B215; Blue, HDM+100B215; Dark blue, HDM+200B215. Data are presented as mean ± SD (# and *, *p* < 0.05, compared to CTR and HDM, respectively).

**Figure 6 nutrients-14-05024-f006:**
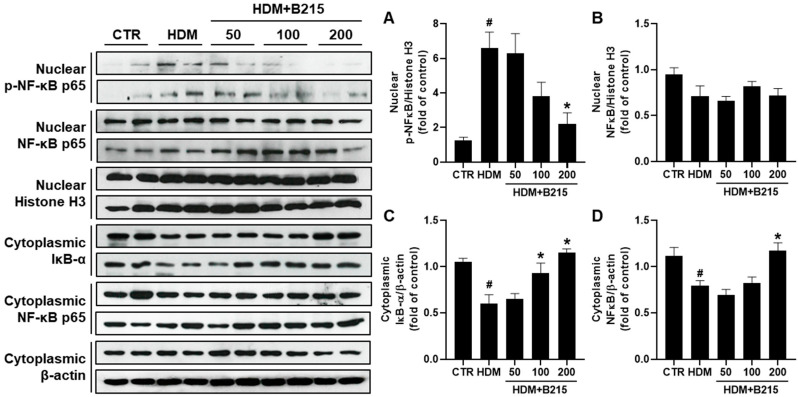
Effect of B215 on NF-κB activation in HDM-induced lung inflammation model. Nucleus was separated from the cytoplasm, and protein was extracted to assess the translocation and phosphorylation of NF-κB and degradation of IκB through immunoblotting. Relative levels of (**A**) phosphorylated NF-κB in nucleus, (**B**) total NF-κB in nucleus, (**C**) IκB-α in cytosol, and (**D**) total NF-κB in cytosol were presented. β-actin and Histone H3 were used as an internal control. Data are presented as mean ± SD (# and *, *p* < 0.05, compared to CTR and HDM, respectively).

**Figure 7 nutrients-14-05024-f007:**
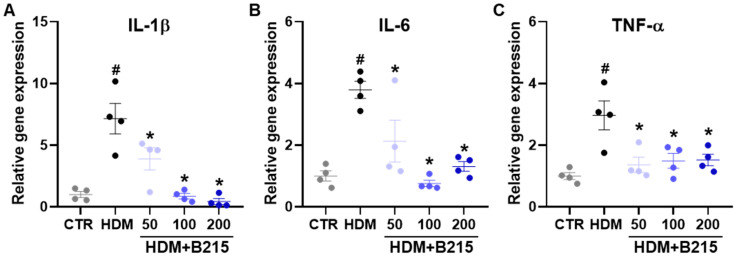
Effect of B215 on pro-inflammatory gene expressions in lungs of HDM-induced mice. Expression of (**A**) IL-1β, (**B**) IL-6, and (**C**) TNF-α were analyzed with real-time qPCR. Gray, Control; Black, HDM; Light blue, HDM+50B215; Blue, HDM+100B215; Dark blue, HDM+200B215. Data are presented as mean ± SD (# and *, *p* < 0.05, compared to CTR and HDM, respectively).

**Table 1 nutrients-14-05024-t001:** Primer lists for real-time quantitative PCR (qPCR).

	Forward (5′ → 3′)	Reverse (5′ → 3′)
*Gapdh* [17]	ATCACCATCTTCCAGGAG	ATGGACTGTGGTCATGAG
*Il1b* [18]	AGTTGACGGACCCCAAAAG	AGCTGGATGCTCTCATCAGG
*Il6* [18]	CCAGGTAGCTATGGTACTCCA	GCTACCAAACTGGCTATAATC
*Tnfa* [19]	CACAGAAGCATGATCCGCGACGT	CGGCAGAGAGGAGGTTGACTTTCT

**Table 2 nutrients-14-05024-t002:** Effect of B215 on plasma biochemical parameters in HDM-induced mice.

	AST (U/L)	ALT (U/L)	TG (mg/dL)	Cholesterol (mg/dL)
CTR (*n* = 8)	64.10 ± 56.72	253.65 ± 79.58	93.13 ± 26.14	103.75 ± 11.11
HDM (*n* = 8)	57.86 ± 24.37	211.80 ± 41.18	67.53 ± 19.01	90.50 ± 10.06
HDM+50B215 (*n* = 8)	48.33 ± 16.93	151.53 ± 99.03	94.64 ± 18.13	102.00 ± 8.72
HDM+100B215 (*n* = 8)	44.04 ± 16.03	124.83 ± 59.32	104.74 ± 20.83 *	102.38 ± 11.88
HDM+200B215 (*n* = 8)	72.21 ± 29.27	241.68 ± 77.48	93.45 ± 19.52	104.38 ± 11.19

Data are presented as mean ± SD (* *p* < 0.05, compared to HDM). AST, aspartate aminotransferase; ALT, alanine aminotransferase; TG, triglyceride.

**Table 3 nutrients-14-05024-t003:** Quantitative analysis of immune cell populations in BALF.

	Neutrophil	Eosinophil	Macrophage	Lymphocyte	Basophil
CTR (*n* = 8)	55,175 ± 8053	22,660 ± 8831	213,955 ± 84,259	437,068 ± 72,351	9345 ± 4441
HDM (*n* = 8)	136,915 ± 43,542 #	177,660 ± 57,119 #	416,829 ± 93,790 #	719,788 ± 187,911 #	58,694 ± 51,335 #
HDM+50B215 (*n* = 8)	63,950 ± 36,434 *	67,800 ± 30,746 *	343,780 ± 136,016	990,873 ± 137,140 *	9814 ± 3381 *
HDM+100B215 (*n* = 8)	63,791 ± 45,781 *	52,811 ± 22,536 *	259,221 ± 83,073 *	978,280 ± 77,775 *	5580 ± 3303 *
HDM+200B215 (*n* = 8)	50,826 ± 43,614 *	56,610 ± 31,207 *	204,960 ± 119,757 *	746,438 ± 152,338	9998 ± 7595 *

Data are presented as mean ± SD (# and *, *p* < 0.05, compared to CTR and HDM, respectively).

## Data Availability

Not applicable.

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
