# Peer review of "Citrus junos Tanaka Peel Extract Ameliorates HDM-Induced Lung Inflammation and Immune Responses In Vivo"

_nutrients, 2022, doi:10.3390/nu14235024_

Round 1

Reviewer 1 Report

Dear authors, 

I applaud the idea of constructing a research model that evaluates possible effects of a substance on both a molecular and macroscopic level. The methods used are well explained and the approach is innovative. I look forward to seeing the potential clinical applications of your research, as mice are but a starting point, however convincing it may be.

Author Response

We sincerely appreciate the reviewer for these generous and positive comments. We hope to our investigation will be a step forward to maintaining lung health.

As suggested, we checked and edited English language through specialized institution of MDPI. We attach the certification of English language editing by MDPI.

Reviewer 2 Report

Dear authors,

The manuscript entitled "Citrus junos Tanaka Peel Extract Ameliorates HDM-induced Lung Inflammation and Immune Responses in vivo” was to investigate the protective effects of Citrus junos Tanaka peel extract (B215) against lung inflammation were examined, and efforts were made to understand the underlying protective mechanism using HDM-induced lung inflammation murine model. It presents scientific relevance for the area of Biochemistry and Medicine area.

After consulting www.sciencedirect.com and https://pubmed.ncbi.nlm.nih.gov/, publications were found for some authors involving the theme. However, you need to change some details:

-I suggest at the end of the introduction, I suggest highlighting the "innovative" proposal of the method, as well as the advantages/disadvantages.

Plant Extracts - Preparation of B215

- line 61-69, “Protocol for obtaining extracts” section: Was this protocol based on any references? If yes, add!

Discussion section:

-I suggest more extensive comments.

Conclusion: Adequate, but I suggest to indicate disadvantages/limitations of the method and the study! Perhaps, to highlight the text in the 'Limits of the study' section.

Author Response

We would like to thank reviewers for spending substantial amount of time looking over the manuscript and for their valuable comments. We are glad that comments and suggestions helped us to improve the quality of the manuscript. We have made necessary corrections as suggested and responded to comments on point-by-point basis.

Reviewer #2

Comments: Extensive editing of English language and style required.

As suggested, we checked and edited English language through specialized institution. We attach the certification of English language editing by MDPI. Please see the attachment.

Comments: I suggest at the end of the introduction, I suggest highlighting the "innovative" proposal of the method, as well as the advantages/disadvantages.

As suggested, we have included the novel study proposal in both introduction and discussion sections of the revised manuscript. Specific texts in the revision are as follows,

Introduction

Generally, asthma patients were treated with corticosteroids and long-acting bronchodilators; nevertheless, these treatments have their own side effects. Moreover, there are no preventive medications for pulmonary inflammation or asthma. Functional foods derived from natural products could be an alternate approach to prevent chronic asthma or lung inflammation, without side effects.

Discussion

Airway inflammation is central to the pathophysiology and clinical expression of asthma. The clinical manifestations of asthma include dyspnea, wheezing, chest tightness, cough, and airway obstruction. In general, asthma is expressed in a range of intensities, from mild to severe but only 5% of asthma cases are severe. The heterogeneity of asthma depends on multiple factors, including lifestyle, age, occupation, allergic condition, history and/or presence of sinusitis. Further, comorbidities such as rhinosinusitis, gastroesophageal reflux disease (GERD), obstructive sleep apnea, obesity, anxiety, and depression increase the risk of frequent exacerbations. Also, these comorbidities are often detrimental to asthma treatment. Traditionally, inhaled corticosteroids (ICS) are widely accepted treatment. However, ICS is linked to systemic adverse effects such as osteoporosis, bone fractures, diabetes, ocular disorders, and respiratory infections. Hence, functional foods derived from natural products could be a safer and more effective treatment for allergic asthma.

(1) Plant Extracts - Preparation of B215 - line 61-69, “Protocol for obtaining extracts” section: Was this protocol based on any references? If yes, add!

As suggested, missing references were inserted.

(2) Discussion section: I suggest more extensive comments.

As suggested, we have added some of the discussion points regarding the novel concept and potential mechanisms involved in this study. We hope our crisp and clear discussion on beneficial effects of B215 on HDM-induced lung inflammation in murine model will enhance the readability and reach out to the wide scientific community.

(3) Conclusion: Adequate, but I suggest to indicate disadvantages/limitations of the method and the study! Perhaps, to highlight the text in the 'Limits of the study' section.

As suggested, we have included limitations of the study under heading limits of the study. Specific texts in the revision are as follows,

Limits of the study

B215 regulatory effect on adaptive immune responses is still unknown. Also, it is unknown which B215 components contributed to the anti-inflammatory effect against HDM-induced lung inflammation. Thus, further investigations are necessary to validate the regulatory effect and the component responsible for the anti-inflammatory effect
